# Increasing Rural Recruitment and Retention through Rural Exposure during Undergraduate Training: An Integrative Review

**DOI:** 10.3390/ijerph17176423

**Published:** 2020-09-03

**Authors:** Jens Holst

**Affiliations:** Department of Nursing and Health Sciences, Fulda University of Applied Sciences, Leipziger Strasse 123, D-36037 Fulda, Germany; jens.holst@pg.hs-fulda.de; Tel.: +49-661-9640-6430

**Keywords:** undergraduate medical training, rural exposure, rural placement, rural practice, recruitment and retention, curriculum, medical workforce, integrative review

## Abstract

*Objectives:* Ensuring nationwide access to medical care challenges health systems worldwide. Rural exposure during undergraduate medical training is promising as a means for overcoming the shortage of physicians outside urban areas, but the effectiveness is widely unknown. This integrative review assesses the effects of rural placements during undergraduate medical training on graduates’ likelihood to take up rural practice. *Methods:* The paper presents the results of a longitudinal review of the literature published in PubMed, Embase, Google Scholar and elsewhere on the measurable effects of rural placements and internships during medical training on the number of graduates in rural practice. *Results:* The combined database and hand search identified 38 suitable primary studies with rather heterogeneous interventions, endpoints and results, mostly cross-sectional and control studies. The analysis of the existing evidence exhibited predominantly positive but rather weak correlations between rural placements during undergraduate medical training and later rural practice. Beyond the initial scope, the review underpinned rural upbringing to be the strongest predictor for rural practice. *Conclusions:* This review confirms that rural exposure during undergraduate medical training to contributes to recruitment and retention in nonurban settings. It can play a role within a broader strategy for overcoming the shortage of rural practitioners. Rural placements during medical education turned out to be particularly effective for rural-entry students. Given the increasing funding being directed towards medical schools to produce graduates that will work rurally, more robust high-quality research is needed.

## 1. Introduction

Safeguarding medical care in rural areas is currently posing growing challenges to healthcare systems around the world, including smaller and relatively densely populated countries. Ever-growing problems in maintaining a sufficient level of nationwide medical care, particularly in rural, remote and structurally weak areas, challenge many countries around the world. Even high absolute numbers of physicians available in a country do not prevent it from experiencing the threat of physician shortages in rural and remote regions.

A long catalogue of measures exists to alleviate the shortage of rural practitioners and promote young medical professionals’ motivation to work in rural areas. It ranges from changes in outpatient or inpatient care delivery to structural cross-sector adjustments, including changes in the roles and tasks of other healthcare professions. Other interventions to recruit and retain medical doctors in rural and remote areas, such as financial incentives, targeted and international recruitment, professional and personal support for physicians working in rural and remote areas, and educational interventions, are regarded as important levers for a sustainable increase in the likelihood of future doctors to work in rural areas. The World Health Organization recommends a series of measures to overcome the shortage of rural practitioners: (1) the targeted selection of students with a rural background, (2) study opportunities outside large hospitals and larger cities, (3) clinical placements and internships in rural areas during undergraduate training, and (4) a stronger orientation of curricula towards general medicine contents and topics which are relevant to rural medical practice, ranging from outreach health promotion to the development of practical skills for medical care under resource-poor conditions [1].

Along with elective courses and some isolated initiatives, even high-income countries are still lagging behind in the endeavour to integrate rural health topics into medical education [2]. However, some countries, especially large states such as Canada, Australia and the USA, where the consequences of rural practitioner shortages are much more dramatic due to the enormous geographical distances, have made better progress. Since the 1990s, these countries have pursued various strategies to generate young professionals for rural medical care through adequate interventions during medical training.

Abundant literature points to the potential of appropriate educational interventions in medical training to increase the likelihood of taking up rural practice after graduation. The evidence, however, is to a large extent based on qualitative studies assessing the effect of general and especially rural medical placements or other rural-oriented curricular interventions in medical training programmes on the priority setting and stated future planning of medical students (e.g., [3,4,5,6,7]). In addition, the vast majority of these studies originate from the initiators of rural placement programmes or other curricular interventions aimed at increasing the motivation of medical graduates to practise in rural regions. The mostly positive assessment of the effectiveness of these interventions in medical training on the prospects of success is therefore likely to be subject to a certain bias.

The objective of this integrative longitudinal analysis was to extend the body of evidence regarding the expected positive effect of rural placements and other curricular interventions during undergraduate medical training on the recruitment or retention of rural practitioners. Therefore, the present review assesses the potential of all types of rural-practice interventions in undergraduate medical education to measurably improve the recruitment and retention of rural practitioners. A systematic analysis of existing studies promises to provide evidence on whether a stronger orientation of medical training towards rural medical practice actually and quantifiably contributes to an increase in the recruitment and retention of physicians in rural and remote areas.

While recruitment focuses on attracting physician and medical students to work in rural positions, retention refers to keeping employed health professionals employed in rural practice, the first being an indispensable condition for the latter. As rural employment is the essential outcome of this review, data about recruitment and retention have to be taken into consideration. Measuring the effectiveness of rural internships during undergraduate medical training needs to count both the number of physicians taking up rural practice and of those working in rural and remote areas. Consequently, the review focuses on the geographical locations of physicians after graduation. The intervention consists of a relatively broad array of rural placements during undergraduate training, ranging from short-term assignments to several years of training in rural areas. Medical graduates who have not taken part in any rural training serve as comparison groups. The exclusive outcome of this literature review is the number or share of medical graduates practicing as generalists or specialists in rural or remote areas; mere expressions of desire or intention by medical students or graduates were not taken into account. To obtain a broader picture, no restrictions were made regarding the study design, the medical specialty, the examination year, the country or the region.

This raises the question as to whether, and to what extent, findings in one country can be transferred to others which have different health and educational systems. However, international health systems comparisons often reveal surprising similarities between basic structures and functionalities of healthcare systems, which tend to exhibit common characteristics beyond the specific features and arrangements at first glance. Hence, the analysis of selected country experiences can provide useful information for policy conclusions and recommendations to be applied by other countries. The objective of this review was to extend the body of evidence regarding the positive effect of rural placements and other curricular interventions in undergraduate medical training on the recruitment and retention of rural practitioners.

## 2. Materials and Methods

This paper is based on a literature search aimed at identifying all types of quantitative research in respect to the impact of undergraduate rural medical training interventions on the factual practice of graduates in rural or remote areas. The primary search was carried out between September and November 2018 in the relevant databases PubMed and Embase, using a combination of the terms “medical AND training”, “rural AND health”, “recruitment” and “retention”, combined with the Boolean operators “OR” and “AND”. In order to obtain a longer-term and integrative picture, no timely, technical, geographical or system-related restrictions were made. An additional search was carried out using Google Scholar as well as the unspecific search engine Google with the same combination of terms. Important additions were provided by an analysis of the relevant literature retrieved from the bibliographic references of the identified studies. The following Table 1 exhibits the target group, the type of intervention, control, and the outcomes examined

The indispensable prerequisite for a study to be included in the review was quantifiable information about the type and location of physicians’ practices after graduation. Only quantitative analyses of the relationship between all types of rural curriculum intervention and content and the actual commencement or exercise of medical practice in a nonurban, structurally weak or remote area allowed conclusions to be drawn about the effectiveness of curricular interventions. This review does not include purely qualitative studies and studies concerning the plans and intentions of students expressed in connection with the content of rural medicine training, unless they also contained quantifiable information on rural medical practice.

Likewise, studies on the effects of rural internships and other curricular interventions in medical education on the frequency of subsequent advanced training or practice in general medicine were not included, unless they also contained information on the number of physicians actually practicing in rural and remote areas. This is worth mentioning as specialisation in general medicine is often regarded as a favourable prerequisite or even as a proxy for rural practice, which primarily requires generalists and fewer specialists. Despite the fact that rural general practitioners play a predominant role in many studies, this review did not restrict the scope to generalists and family doctors as some studies also included the effect of nonurban placements on the rural location of specialists.

The initial search with the above-mentioned terms in pertinent databases delivered 276 hits (171 in PubMed, 105 in Embase), of which 61 hits appeared to be primarily suitable according to the headings. In addition, the Google Scholar search and the additional manual search in the unspecific search engine brought 21 more publications to light that had not been detected in the database searches. The review also encompassed repeated direct online searches for papers not included in the literature databases, targeted online searches for additional articles by authors who were frequently identified in the database search, and a review of the bibliographies of the studies and reports retrieved. The screening of the respective reference lists pointed to a further 72 studies that promised to be relevant in answering the research question. After reviewing the titles and abstracts, 49 (31.8%) of the total of 154 publications were excluded as they did not fulfil the study criteria, and another 67 studies were excluded for the same reason after reviewing the full texts (43.5%). The reasons for exclusion were as follows:Lack of quantitative information on physicians practicing in rural areas after graduation,Restriction to student opinions and declarations of intent,Provision of purely qualitative information on various aspects of rural tracks,Information on the effectiveness of postgraduate interventions,Reference to outcomes other than rural practice,Mere expressions of opinions and expert perceptions.

As shown in the flow diagram below (Figure 1), 38 primary studies (16 from the database search, 19 from the review of bibliographies and 3 from the hand search) were finally included in the literature review. The largest number of primary studies, published between 1987 and 2018, stem from Australia (14), followed by the United States of America (13), Canada (7), Norway (2), Japan and the Philippines (1 each). The majority of the hits were published in English, in addition to one French and one Norwegian article, which could be found due to the English abstracts (a detailed presentation of the studies included can be found in Appendix A.

The assessment of the study quality was performed by the author, taking into consideration the clarity of the research question and objective, specification of the study population, consistency of the target population, comprehensibility of the information on inclusion and exclusion criteria, appropriateness of the time frame of the survey, determination of the exposure measures or levels, clarity and reliability of the information on outcome measures and the recording of possible confounders. Due to a general lack of information on the participation rates, strength or variance and effect estimates or dropouts, particularly in older studies, these parameters could not be strictly applied in the comparative quality analysis in order to not lose potentially interesting results.

## 3. Results

All in all, the primary studies exhibit a pronounced heterogeneity, particularly in terms of the size of the study populations and the type and intensity of interventions, ranging from short rural assignments to complete rural medical training in a geographically remote school. Moreover, the objectives associated with curricular interventions in undergraduate medical education and, above all, the endpoints examined differ considerably among the studies included in the review, of which 15 (39.5%) were control studies and cross-sectional studies, respectively; nine were database-driven and six survey-based. Moreover, the review includes five predominantly survey-based longitudinal studies (13.2%), one quasi-experimental study (2.6%) and two mixed-methods studies (5.3%).

A total of 23 of the 38 studies compared the share of rural medical practitioners between medical students who had taken part in some type of rural internship during undergraduate training and those who had not participated in such interventions. Rural exposure during medical education increased the likelihood of later rural practice by more than four times (4.2) on average, but with a wide variation ranging between 1.34 and 19.1 (standard deviation 3.92) [8,9,10,11,12,13,14,15,16,17,18,19,20,21,22,23,24,25,26,27,28].

One study also investigated the effect of interventions in different phases of medical education and found rural internships and tracks towards the end of undergraduate training to be particularly effective (the likelihood of later rural practice increased by 1.5 after internships in the third year, and by 3.0 in the fifth year) [29]. Another analysis showed a clear correlation of postgraduate rural practice with the length of medical training in a rural environment: Compared to 1 to 2 years, 3 to 4 years of training in rural and remote medical schools increased the likelihood of becoming a rural practitioner by 9.3 instead of 2.5 times [30]. Thus, despite the variability of the findings, it can be assumed that rural curriculum interventions in undergraduate medical training have the potential to favour later medical practice in rural and remote areas; this correlation is rather weak for some interventions but relatively strong for others.

Ten studies recorded the share of rural practitioners after medical training outside larger cities and urban medical schools. The results were also quite heterogeneous, because three studies showed a share of less than 50% [31,32,33], five studies slightly above 50% [34,35,36,37,38,39], one study did not explicitly quantify the proportion [40], and only one study exhibited a very large share of medical graduates whose undergraduate training included rural practice interventions and who took up jobs in rural areas [41]. The average share was 51.3%, with a standard deviation of 17.5%. Despite the heterogeneity of the results, the proportion of rural practitioners following targeted curricular interventions tends to be higher than after conventional medical education.

Five studies determined the probability of taking up rural practice after tracks, internships and other training opportunities outside cities and urban medical schools, mainly by means of odds ratios, which were 2.12 (*p* = 0.15) [42], 2.6 (95% CI = 2.1–3.4) [43], 6.4 [44] and 8.4 (95% CI = 2.1–33.5, *p* = 0.002) [32], or the prevalence odds ratio, which was 3.9 [45]. Here, too, the obvious variability of the results has to be stressed. A further study assessed the effectiveness of rural exposure during undergraduate medical training in relation to urban or rural origin of students and found that an internship outside larger cities can even outweigh the aversion of urban-entry students to rural practice [46].

While two of the studies provided evidence for an intuitively comprehensible dose-response relationship [47] between the length and intensity of rural exposure during undergraduate medical training and the likelihood to work as rural practitioner [29,30], the empirical basis for this assumption remains rather weak. A number of studies provide evidence that, in particular, longer internships and rotations during medical training correlate with positive rural workforce outcomes [11,30,32,48]. It should be noted, however, that longer periods of rural exposure during undergraduate training were often followed by additional rural training and postgraduate experience. This makes it impossible to separate the positive correlation properly from the effects of postgraduate rural practice [49].

## 4. Discussion

Overcoming the growing medical workforce shortages in rural areas of many countries worldwide requires effective and sustainable strategies. Increasing the number of medical practitioners in rural and remote areas through medical training interventions is a complex health-policy challenge that calls for a broad range of intelligent and coordinated measures. Universities and medical schools can make a significant contribution.

One step would be the further propagation and upgrading of academic general practice and family medicine, particularly in those countries where generalists do not have the same reputation as other clinical specialists [1]. In addition to a more practical orientation of undergraduate medical education, strengthening general medicine is considered a relevant prerequisite for improving healthcare in rural areas [50]. This review shows that the combination of the two approaches is promising, particularly a stronger practical orientation in the form of rural tracks, which in almost all included studies focused on general and family medicine. Extending the availability and intensity of rural exposure in undergraduate medical education ultimately requires general medicine chairs to receive the same rights, possibilities and perspectives as their colleagues in other clinical disciplines. In order to make general medicine more practice-oriented and thereby more attractive, providing care to patients must be or become an important part of the scope of tasks in academic general medicine. This will hardly work as long as general and family medicine is not a part of the standard healthcare provided by academic medical schools, and as long as university general practice lecturers cannot bill for medical services in the same way as their colleagues in other specialisation fields.

This review identified and assessed a critical number of studies on the effectiveness of rural exposure during undergraduate medical training in increasing the rural workforce. In particular, longer and more intensive rural placements of medical students turned out to be promising in contributing to a moderate or even significant increase in the number of graduates working in rural practice. This assessment is consistent with other meta-analyses and reviews [50,51,52,53] showing that particularly larger programmes, which combine several approaches and measures, lead to a higher share of medical graduates taking up employment in rural areas [54,55].

The strongest effect can be observed if a relevant part or the whole medical training take place at a rural or remote medical school, i.e., outside metropolitan and large cities. However, findings are inconsistent here as well [51,56,57], which is probably due, among other things, to the diversity of the interventions. Understandably, the outcome of rural placements and tracks during medical training depends not only on the extent, but also on the nature and quality of rural exposure [58]. Despite this, some findings question the connection between positive experiences of students acquired during rural medical placements or internships and subsequent rural practice [59]. Further research is needed to determine whether and under what conditions longer or more intensive internships and rotations during undergraduate medical education can effectively contribute to increasing the rural medical workforce.

Beyond the actual research question, this review revealed another finding, which is worth mentioning in this context as it is highly important for the evaluation of the effectiveness of rural interventions and exposures during medical training: the most promising strategy for promoting employment in rural practice after graduation seems to be the targeted selection of students of rural origin who are more likely than others to end up working in rural areas. A considerable share of studies included in this review show that the rural upbringing of medical students or their partners turns out to be the strongest predictor for rural medical practice [13,16,17,18,20,21,23,26,28,37,38,41,42,43]. This finding is consistent with a large number of earlier observations and findings [1,52,55,57,60,61,62,63,64]. Physicians who have grown up or at least spent a certain time of their lives in the countryside or in small towns are much more likely to take up a rural practice than their colleagues with an urban upbringing. In addition, rural medical training tracks and internships had the greatest effect on future medical students with a rural background [21].

Hence, prioritising the allocation of medical school places to students of rural origin might be a promising approach for training physicians who are willing to take up rural medical practice after graduation. It can be assumed that an appropriate adaptation of the framework conditions and admission criteria for medical school assignments can increase the effectiveness of internships in rural medicine and thereby contribute to reducing the undersupply of rural medical workforce. However, such a policy has to take into account the trade-off between individual freedom and regulatory interventions, and it might create a series of concerns regarding fairness, individual priority setting and confidentiality of data, among others. Moreover, the inclusion of voluntary or possibly mandatory rural exposure, i.e., outside university clinics, teaching hospitals and urban practices, in the curricula of medical schools can contribute to an increase in the share of graduates taking up rural medical practice [65,66]. The suspicion that more and prolonged rural internships and placements in undergraduate medical training could negatively affect the level of performance proves to be unfounded [15,41,67].

## 5. Limitations

The present analysis of the existing evidence regarding the impact of rural interventions during undergraduate medical training on the recruitment and retention of rural practitioners has several limitations. First of all, the availability of studies with good and reliable empirical evidence is rather low. Although some studies of higher quality have been published in recent years, the evidence for rural exposure during undergraduate medical training having contributed to overcoming the shortage of rural practitioners is relatively weak. Another fundamental limitation of this review results from the conceptual and methodological heterogeneity of many primary studies included, which exhibit important differences with regard to both the interventions and endpoints. Hence, a summary assessment of the existing evidence was only possible according to the quite-general criteria and without sufficient consideration of any special features in the respective countries and health systems.

Moreover, only a small number of studies systematically consider potential confounders that may have influenced the decision of graduates to take up rural practice independently of the intervention observed in the respective study. These may be individual factors such as rural upbringing and family or social ties, framework conditions such as additional incentives, for example in the form of grants and special remuneration opportunities, support in setting up and leading practices, or special training opportunities.

It has to be mentioned that this review does not address the difference that exists between recruitment and retention, although the barriers to long-term employment are certainly higher than those for taking up rural practice. This limitation derives from the fact that the studies identified did not exhibit differentiated data regarding recruitment and retention—they captured rural practice at a given point in time, and not its duration. Hence, it was impossible to draw specific conclusions about both rural recruitment and retention.

Finally, one should bear in mind that the results from one country cannot be simply transferred to other countries. Different training, financing and healthcare systems coincide with other incentive and control measures to increase the share of rural physicians, and different (health) policy frameworks limit the generalisability of the findings described above. However, due to the fundamentally similar trend of the study results, regardless of region, system and other influencing factors, the transferability of some country-specific findings can be assumed [52,68]. Different framework conditions regarding the centralisation or decentralisation of higher education and country-specific political, normative, financial and administrative arrangements have rather indirect effects on the relationship between curricular interventions and outcomes. Moreover, they do not call into question the general relationships presented in the review, especially as the impedimental causes for increasing the medical workforce are in principle very similar in all countries [1,63,69].

## 6. Conclusions

The present review contributes to the existing literature as it provides a longitudinal overview of the outcome of a broad array of attempts and approaches to promote rural practice through interventions in undergraduate medical training in a number of countries around the world. The global focus does not only reconfirm the potential of rural exposure during undergraduate medical training in contributing to recruitment and thereby also to retention in nonurban settings, it also shows the generally increasing effectiveness of such interventions over time. Regardless of the country and healthcare system in place, rural placements during medical training have some potential to support the endeavour to reduce the shortage of rural physicians. Along with the control of supply and demand, financial incentives and other measures, undergraduate medical education can play a considerable role in the recruitment and later retention of rural physicians. Beyond the underlying research question in the narrower sense, the review corroborates existing evidence for the preferential admission of students with a rural upbringing to medical schools as being an important component of a successful strategy to maintain and stabilise the medical workforce in nonurban and even remote regions. Medical education outside larger cities and highly specialised university hospitals, which familiarises students with special features of rural healthcare, is promising to promote graduates to take up rural practice—this applies particularly to rural-entry students. In addition, more intensive, mandatory contact with healthcare outside larger cities and metropolitan areas can contribute to the overall upgrade of general practice and family medicine, and especially rural medicine. There is a need for further research to clarify which interventions are most successful in increasing the likelihood of prospective medical graduates to work in rural practices, and which additional factors tend to be promotive or inhibitory for rural practice.

## Figures and Tables

**Figure 1 ijerph-17-06423-f001:**
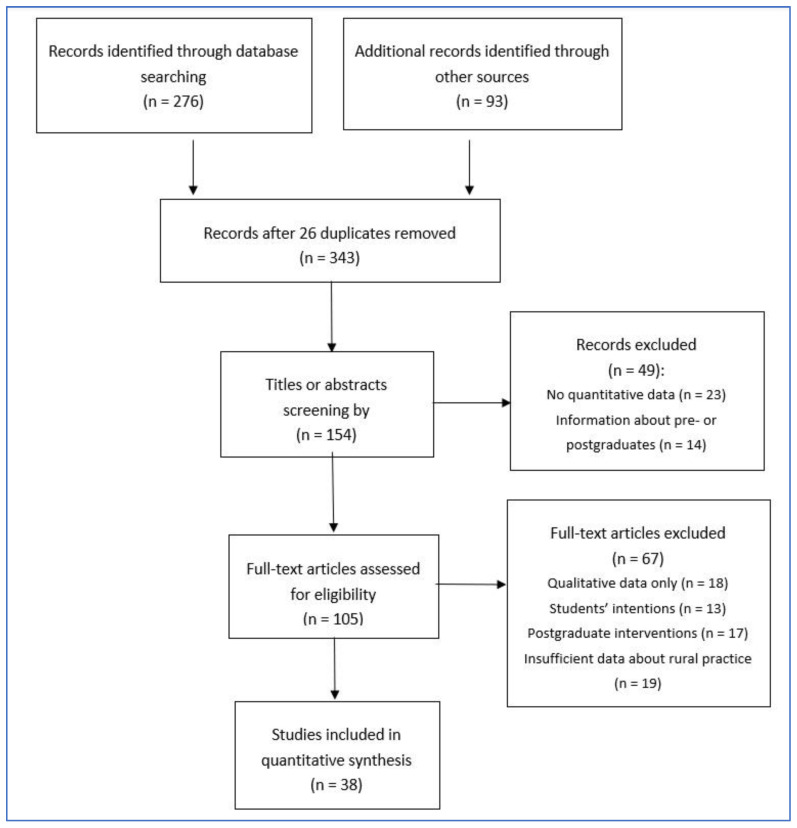
Flow diagram of study selection.

**Table 1 ijerph-17-06423-t001:** PICO scheme.

**P**opulation	Medical students
**I**ntervention	All types of rural placements and internships during undergraduate training
**C**omparator	No rural exposure during undergraduate training
**O**utcome	Rural employment after graduation

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
