# Peer review of "Increasing Rural Recruitment and Retention through Rural Exposure during Undergraduate Training: An Integrative Review"

_ijerph, 2020, doi:10.3390/ijerph17176423_

Round 1
Reviewer 1 Report
Feedback to Author
While this paper could be useful to further inform the literature on this particular topic area. I find that this reported systematic review lacks rigor and does not appear to follow (or articulate that it has followed) the PRISMA guidelines.
The see the following as major issues:
Key aspects of the PRIMSA checklist are not clearly addressed in the reporting of the methods e.g. where is the clear PICO/S statement. I do not consider the current objective (on lines 84-87) to be clear or precise enough for a systematic review. Also it is stated differently elsewhere (see lines 13-14).
It is standard practice for quality systematic review to have a priori protocol published online prior to undertaking the review, if there is/was one it needs to be registered (e.g. PROSPERO, Open Science) and stated in the methods.
There needs to be clear definitions of key terms and use of MESH headings to ensure clarity of the topic and suitable terms used in the search. This is particularly important for a search across many countries with varied terminology being used.
For the review to only identify 276 articles on its initial search suggests to me that a thorough and systematic search of the literature has not been undertaken with appropriate search terms and database searching processes. As a direct comparison (ref 53) identified 392 articles with a focus on rural workforce outcomes from undergraduate medical programs Australian only. There is also no indication that there were other researchers involved in the title and abstract and full text screening processes. This is a critical step in the systematic review process that needs to clearly undertaken and stated.
i am unclear what quality assessment tool was used to assess the identified papers.
Methods lines 107-111 would need referencing to justify statements made.
The search was undertaken in in Sept-October 2018 and there is no indication of a repeat of this process to capture more recent publications, hence it is not a review of all current work.
I have only highlighted issues with the methods as these are key to any systematic review being published. Hence issues in the rest of the document are not raised given there is not enough clarity in the methods to ensure that the review is systematic and unbiased with adequate rigor to the processes undertaken.
Author Response
Thank you very much for your review and the valuable comments and suggestions.
The PRISMA criteria were applied in the review although they are not directly mentioned one by one; for making the approach clearer, a PICOS table was included.
The objective was presented more precisely and inconsistencies overcome (lines 20-21 and 73-80).
An à priori was not published since the review from a research project covering various aspects of undergraduate training and was destined stepwise from the broader initial approach. Hence a study protocol was not developed early enough for be published in advance. Due to the development of the review, the screening process relied on the author (line 345).
Various attempts were made to search with MESH terms. However, unfortunately the search with MESH terms turned out to be unsatisfactory and did not allow a targeted search of pertinent studies without hitting on a large share comments, opinion pieces and merely qualitative studies. The inclusion of terms related to the study type (such as control or cross-sectional studies) did not lead to better targeting since particularly older research could not be identified. The combination of search terms with the Bool variables AND and OR resulted in either too many and very few or even null hits (cf. lines 122/123).
Criteria of the study quality assessment are added (lines 345-352).
Additional search for more recent publications is currently unfeasible; anyhow, a quick overview does not point to a relevant number of new research on this topic.
Several parts of the paper were reformulated in order to make the various points clearer and point out the results and limitations. However, due to the initial planning and development of the review, some issues cannot be fully sorted out without starting a completely new approach. Nonetheless the author hopes that the paper is now suitable for publication. Please find attached the amended version with track changes.

Reviewer 2 Report
The aim of the manuscript titled Increasing Rural Recruitment and Retention through Rural Exposure during Undergraduate Training: A Systematic Review is to assess the potential of rural interventions in undergraduate medical education to contribute to improving the recruitment and retention of rural practitioners.
In order to achieve this objective, the author develops a systematic literature review with meta-analysis of the existing literature on quantitative results of undergraduate rural medical training interventions on the factual practice of graduates in rural or remote areas. The search was carried in the databases PubMed and Embase, complemented with searches in Google Scholar and Google using the combination of the terms ‘medical training’, ´rural health’, ‘recruitment’ and ‘retention’ and without temporal, technical, geographical or system-related restrictions.
The paper addresses a relevant topic from a public health perspective. In fact, human resources for health is the most relevant challenge from the organisation of the response to people’s health needs and, within it, the physician shortages in rural and remote regions is probably the problem most difficult for countries to deal with.
The paper is well organised and very well written, which facilitates the understanding of the contents that are developed throughout the manuscript. The abstract reflects the content of the article. There is consistency among the different sections of the manuscript.
The title of the manuscript appropriately describes the content of the paper. The Introduction section is well achieved in my opinion since it provides enough elements for the reader to perceive the importance of the recruitment and retention of rural practitioners as a problem, on one hand, and the complexity of the responses to the problem, on the other. The section is successful in capturing the reader’s interest. The author also modulates the reader’s expectations when he explicitly states the objective of the manuscript.
The description of the methods used - a systematic literature review with meta-analysis- is well developed and clear. The path from the initial 343 records (after removal of duplicated records) to the final 38 studies included in the meta-analysis is well explained. The Prisma flow diagram is very useful in helping understand the rationale of the search (the f letter is missing in the word ‘flow’ in the figure’s title).
In the results section, table 1 is well structured to present an appropriate synthesis of the findings coming from the 38 studies that were analysed.
The discussion is pertinent to the aim of the study and offers a good analysis in the context of the current state of the art providing pertinent references.
The conclusions, correctly provided at the end of discussion section, are correctly presented, highlighting the limitations of the meta-analysis developed, especially the reflexions on the strengths and weaknesses of the transferability of some of the findings.
Author Response
Thank you very much for your review and the positive assessment!
Based on other reviewers comments and recommendations, the author has revised, partly accomplished and proofread the paper after the first round of reviews. Please find attached the file.

Reviewer 3 Report
I have two significant concerns about this paper:
- Table 1 comprises 20 pages of a 32 page paper. This is almost 2/3 of the paper. I would encourage reworking this table to make it more concise and more a relevant summary (as opposed to a 'list') of results of the systematic review.
- I am concerned with the recommendation (p. 27) that students with rural backgrounds be 'targeted' for rural placements. This is problematic on many levels and the discussion, while suggesting this may be an effective strategy, should discuss the problems associated with such targeting.
I also have a few minor concerns:
1. Overall, the paper is well-written, however there is a need for some minor editing.
2. I would encourage the authors to emphasize more strongly that it is widely understood that rural background and rural exposure are indicative of successful recruitment to rural practice (include some references - The Australian Journal of Rural Issues, as well as this journal can offer these). This systematic review confirms the (largely) qualitative data. Important, but not ground-breaking.
3. A major limitation of this meta-analysis concerns the issue of retention. It is fairly well understood that rural background is more predictive of rural recruitment. However, there is little research regarding the association between rural background, recruitment to rural practice, and long term retention in rural practice. This is an important consideration. Could the author speak to this issue?
Author Response
Thank you very much for your valuable comments. Please find below my replies and comments:
The concern regarding the results tables is left to the editors - it can be published online only. Here I wait for recommendations and requirements of the editors.
The recommendation to target students with rural upbringing is slightly attenuated and accompanied by a reference to the concerns such a strategy can raise, mentioning some of them(lines 20-21, 689-691, 1014-1026: Please note that my computer presents corrupted numbers, here I refer to the para starting with "Hence, prioritizing the allocation ..." ); a more in-depth discussion would go beyond the scope of this paper.
For the purpose of this review, the distinction between recruitment and retention lacks relevance since the available evidence is restricted to the number of physicians working in rural areas bur cannot assess the time of recruitment or the length of retention. Additional explanations have been included (lines 81-87) but a detailed discussion about the impact of rural placements non recruitment on the one hand and retention on the other hand cannot be deducted from the available studies.
The paper has been re-edited in order to improve clarity and understanding.
Looking forward and kind regards.

Round 2
Reviewer 1 Report
This manuscript still lacks the rigorous and comprehensive systematic review process I would expect for publication. The following issues have not been addressed: current literature from October 2018 onwards, no second reviewer used for screening, full text review or data extraction. Given the heterogeneity of the studies meta-analysis is perhaps not warranted and there is no actual data provided for the reported meta-analysis (i.e. no forest plot) to support the claim, on lines 18-20.
Author Response
The criticized aspects cannot be compensated retrospectively. Therefore, the formal claim of the article was adjusted.
Reviewer 3 Report
I remain concerned about a Table that comprises almost 2/3 of the manuscript and have encouraged the Editors to provide you with advice if they are interested in publication.
Given your acknowledgement of previous literature that demonstrates the relevance of previous rural exposure to rural recruitment, the contribution your review makes is not clear - especially given the problematic assertion that students with rural backgrounds should be 'targeted' for rural placements.
You need to more clearly clarify the contribution this review makes to the literature.
Your assertion that retention is 'outside the scope' of this study is a fair one, but suggests a limited grasp of rural workforce issues. There is considerable literature in this and other journals noting that rural recruitment is not as difficult as rural retention. Not addressing this should at least be noted as a limitation of the study and an area for future research (i.e. does rural background + rural placement = enhanced retention of rural practitioners?).
Author Response
The table had unintentionally moved in the centre of the paper during the submission or revision process. but was planned as annex. This has been corrected, and three columns were deleted in order to make the table clearer.
The contribution of the review has been made more explicit (lines 343-349).
The reason, why the review does not differ between recruitment and retention was added under the section "Limitations" (lines 399-404).